# Replacing Sedentary Time with Physically Active Behaviour Predicts Improved Body Composition and Metabolic Health Outcomes

**DOI:** 10.3390/ijerph19148760

**Published:** 2022-07-19

**Authors:** Wendy J. O’Brien, Erica L. Rauff, Sarah P. Shultz, McLean Sloughter, Philip W. Fink, Bernhard Breier, Rozanne Kruger

**Affiliations:** 1School of Sport, Exercise and Nutrition, Massey University, Auckland 0632, New Zealand; w.j.obrien@massey.ac.nz (W.J.O.); shultzsarah@seattleu.edu (S.P.S.); philipwilliam.fink@sorbonne-paris-nord.fr (P.W.F.); breier@xtra.co.nz (B.B.); r.kruger@massey.ac.nz (R.K.); 2Kinesiology Department, Seattle University, Seattle, WA 98122, USA; 3Math Department, Seattle University, Seattle, WA 98122, USA; sloughtj@seattleu.edu; 4Department of Sciences and Techniques of Physical and Sports Activities (STAPS), Université Sorbonne Paris Nord, 75013 Paris, France; 5Insitut Biomécanique Humaine Georges Charpak, Université Sorbonne Paris Nord, 75013 Paris, France

**Keywords:** physical activity, accelerometry, exercise, guidelines and recommendations, metabolic health, obesity

## Abstract

Background: Discretionary leisure time for health-promoting physical activity (PA) is limited. This study aimed to predict body composition and metabolic health marker changes from PA reallocation using isotemporal substitution analysis. Methods: Healthy New Zealand women (*n* = 175; 16–45 y) with high BMI (≥25 kg/m^2^) and high body fat percentage (≥30%) were divided into three groups by ethnicity (Māori *n* = 37, Pacific *n* = 54, and New Zealand European *n* = 84). PA, fat mass, lean mass, and metabolic health were assessed. Isotemporal substitution paradigms reallocated 30 min/day of sedentary behaviour to varying PA intensities. Results: Reallocating sedentary behaviour with moderate intensity, PA predicted Māori women would have improved body fat% (14.83%), android fat% (10.74%), and insulin levels (55.27%) while the model predicted Pacific women would have improved waist-to-hip (6.40%) and android-to-gynoid (19.48%) ratios. Replacing sedentary time with moderate-vigorous PA predicted Māori women to have improved BMI (15.33%), waist circumference (9.98%), body fat% (16.16%), android fat% (12.54%), gynoid fat% (10.04%), insulin (55.58%), and leptin (43.86%) levels; for Pacific women, improvement of waist-to-hip-ratio (5.30%) was predicted. Conclusions: Sedentary behaviour must be substituted with PA of at least moderate intensity to reap benefits. Māori women received the greatest benefits when reallocating PA. PA recommendations to improve health should reflect the needs and current activity levels of specific populations.

## 1. Introduction

Ethnic differences in body mass index (BMI), body fat percentage (BF%), and fat distribution have been reported repeatedly [1,2,3,4,5,6], including among Hispanic, African-American, and Caucasian women in the United States [1,2], and in Māori, Pacific, and European populations in New Zealand [3,4]. Specifically, for any given BMI value, European women have a higher BF% than Māori or Pacific women [3]. Large ethnic disparities also exist in obesity prevalence among New Zealand women. Overall, 66% of all New Zealand women have a BMI above the healthy range (i.e., ≥25 kg/m^2^), yet overweight/obesity prevalence varies between ethnic groups, ranging from 91% (Pacific) to 49% (Asian) [7]. Subsequently, similar ethnic disparities are evident in obesity-related disease risk and prevalence in the New Zealand population. For example, type 2 diabetes prevalence is substantially higher among Pacific (9.5%) than Māori (6.8%) or European (3.9%) women, mirroring the ranking of obesity prevalence among these groups [7].

Physical activity (PA) is known to effectively reduce disease risk and improve long-term health and weight management outcomes [8]. Women meeting PA guidelines had a 14% lower risk of coronary heart disease [9] and a 33% reduced risk of type 2 diabetes [10] relative to inactive women; nevertheless, 36% of New Zealand women fail to meet the PA guidelines (compared to 32% and 11%, respectively, among men) [7]. PA participation also differs by ethnicity and follows the ethnic pattern of obesity and its related diseases; 19% of Pacific women complete less than 30 min of PA per week, whilst 15% of Māori and 13% of New Zealand European women do so [7]. However, recommendations for PA duration and intensity (e.g., 30 min/day at moderate intensity) do not clarify what activity (and its subsequent intensity) the recommended PA should replace. This discrepancy is especially important since discretionary leisure time is finite; engaging in one activity prevents using that same period of time to perform an alternative activity. Therefore, identifying which activities provide the greatest benefit to overall health is critical. For example, 30 min/day spent walking instead of watching television would have quite different effects on metabolic health markers than if already active behaviour (e.g., playing basketball) was replaced with walking. To account for the finite nature of available leisure time, analysis to substitute time between exercise intensities may be used to predict the most beneficial volume and intensity of PA on indicators of disease risk and health [11].

Isotemporal substitution analysis is used to predict the cardiometabolic health benefits of a more physically active lifestyle by reallocating time from one activity (e.g., sitting) to another (e.g., stepping) [12,13]. However, this technique has not been employed to examine changes in detailed body composition variables, such as, total and regional lean and fat mass with the reallocation of activities. Excess fat mass, regardless of BMI, has negative consequences on metabolic health [14]. The specific location of fat is also important. Excessive fat in the android region poses a greater risk of inflammation and chronic diseases (e.g., type 2 diabetes, cardiovascular disease) than equally excessive gynoid fat [15]. Understanding the effects of increased PA on such measures of obesity is of increasing public health importance in many countries [16], and in New Zealand in particular, where obesity prevalence ranks fourth in the OECD [17,18].

Given the worldwide epidemic of obesity and its associated complications, understanding the potential health benefits of specific increases in PA is of significant public health importance and could contribute to the implementation of effective health-improvement initiatives. Therefore, the aim of this study was to evaluate the potential benefits of replacing sedentary behaviour with more physically active behaviours on body composition and metabolic health markers in New Zealand women of varying ethnicities with overweight/obesity.

## 2. Methods

This cross-sectional study reports objectively measured PA, body composition, and metabolic data obtained from a subsample of participants within the women’s EXPLORE study [19] conducted in Auckland, New Zealand. The study was approved by the Massey University Human Ethics Committee: Southern A, Reference No.13/13, and conducted in accordance with the Declaration of Helsinki. Prior to data collection, written informed consent was obtained from participants.

### 2.1. Participants

Participants were a subsample from the EXPLORE study, which recruited 406 women aged 16–45 years of Māori (indigenous people of New Zealand), Pacific, or European descent. Participants in this analysis (*n* = 206) were classified as overweight or obese (Body Mass Index, BMI, of ≥25 kg/m^2^) [20] and high body fat (percentage of total body fat, BF%, of ≥30.0%) [21]. Inclusion criteria included being post-menarche but pre-menopausal and having no diagnosed metabolic or other chronic diseases. Ethnicity was self-identified by at least one parent of the ethnicity. Participants were excluded if they were pregnant or lactating.

### 2.2. Physical Activity

Tri-axial accelerometers (Actigraph wGT3X+, Pensacola, FL, USA) were used to assess PA over seven consecutive days. Accelerometers sampled at 100 Hz frequency, and data were downloaded using the low-frequency extension in 60 second epochs. Accelerometers were worn on an elastic belt over the participants’ right hip at all times (excluding water-based activities) during a typical week. To be considered for final analysis, participants must have returned the accelerometer data for ≥10 h/day for ≥4 weeks and/or weekend days [22]. Valid data were returned from 175 participants; 31 participants returned either insufficient (*n* = 19) or no (device malfunction, *n* = 1; device not returned, *n* = 9; device not worn, *n* = 2) data.

Non-wear and sleep time were determined and removed using a publicly available algorithm [23]. Non-wear time was defined as intervals ≥ 60 consecutive min of 0 counts per minute (cpm), with the exception of 1–2 min of activity during that time. Algorithms were run using MATLAB (R2011b 7.13.0.564, The MathWorks Inc., Natick, MA, USA) computer software. Widely used and validated cpm were used to identify PA levels: sedentary (0–99), light (100–2019), moderate (2020–5998), vigorous (≥5999), and moderate to vigorous (MVPA; ≥2020)PA [24].

### 2.3. Anthropometry and Body Composition

The anthropometric assessment was conducted using the International Society for the Advancement of Kinanthropometry protocols [25]. Briefly, waist and hip circumferences, and (standing) height were measured to the nearest 0.1 cm. Body mass (kg) was assessed using the electronic scales incorporated into the air displacement plethysmography device (BodPod, 2007A, Life Measurement Inc., Concord, CA, USA; manufacturer software V4.2+). BMI (kg/m^2^) was calculated from measured height and body mass. Air displacement plethysmography also assessed total BF% using measured thoracic volume and the Siri equation [26]. Regional (android, gynoid) fat and total and regional lean mass were determined from whole-body dual-energy X-ray absorptiometry (Hologic QDR Discovery A with APEX v3.2 software; Hologic Inc., Bedford, MA, USA).

### 2.4. Metabolic Biomarkers

Fasting venous blood samples were drawn between 7:00 and 9:30 am into EDTA and serum vacutainer tubes. An aliquot of EDTA whole blood was immediately frozen at −80 °C for later HbA1c analysis using high-performance liquid chromatography (Biorad Variant Instrumentation, Hercules, CA, USA). The remaining blood was centrifuged at 3500× *g* rpm for 15 min at 4 °C, to obtain plasma and serum samples. Samples were aliquoted and frozen at −80 °C for later analysis. Serum insulin was measured using ADVIA Centaur immunoassay kits (Siemens Healthcare Diagnostics, Munich, Germany). Serum cholesterol, high-density lipoprotein cholesterol (HDL-C), glucose, and triglycerides were measured using automated Dimension Vista procedures (Siemens Healthcare Diagnostics). Cholesterol to HDL-C ratio and low-density lipoprotein cholesterol (LDL-C) were calculated from measured variables. Blood pressure was measured twice using an automated blood pressure monitor (Riester Ri Champion, Rudolf Riester GmbH, Jungingen, Germany) after 5 min of seated rest and the average of the two measurements was recorded.

### 2.5. Dietary Analysis

Total energy intake was calculated from a validated online semi-quantitative food frequency questionnaire [27] analysed using Foodworks Professional 7 (Xyris Software, Sydney, Australia; New Zealand Food Composition Database [28]) for use as a covariate.

### 2.6. Statistical Analysis

Descriptive statistical analyses were carried out using SPSS Statistics 22 for Windows (SPSS, Inc., Chicago, IL, USA). Normality of data was confirmed using histograms and Kolmogorov–Smirnov tests. Data are reported as mean ± SD. One-way ANOVA was used to compare PA, body composition, and metabolic data between ethnic groups. The level of significance was set at *p* < 0.05 for all analyses.

Linear regression was used to examine associations between metabolic markers and body composition variables, and 30 min time blocks of sedentary behaviour, light and moderate PA, and MVPA. Regression models were fit using R 4.1.2 (R Core Team & R Foundation for Statistical Computing, Lucent Technologies, Murray Hill, NJ, USA). No vigorous PA was performed by 55% of women, contradicting the isotemporal model’s assumptions. Hence, vigorous PA was not included in the models. Time blocks of 30 min/day were chosen to roughly align with PA guidelines (150 min/week, commonly interpreted as 30 min/day on 5 days/week) [16]. Isotemporal substitution models were used to estimate the effects of substituting 30 min/day of sedentary behaviour with 30 min/day of any other level of activity, whilst holding total wear-time constant. All-time reallocation is 30 min/day of the stated PA intensity. For each outcome variable in this isotemporal substitution model, the total wear-time variable plus all activities, except the activity of interest, were entered into the model simultaneously, thus time was constrained (i.e., isotemporal). Predicted percentage changes in outcome variables were determined using the difference between predicted and measured means of each variable, divided by the measured mean of the variable, multiplied by 100. Age and total energy intake were covariates in all models; BMI was also included as a covariate for metabolic variables. Power calculations were conducted to give a minimum of 37 participants in each group, to detect an effect size of 0.58 with 80% power. At this number of participants, correlations of 0.44 were able to be detected with 80% power at a significance level of *p* < 0.05.

## 3. Results

A total of 175 participants (*n* = 37 Māori; *n* = 54 Pacific; *n* = 84 European) were included in the current analysis. Metabolic health markers and associated healthy reference ranges are presented in Table 1. Time spent in sedentary behaviour and different intensities of PA differed across the ethnic groups (Table 2). Pacific women spent significantly more time in sedentary behaviour (490 ± 67 min/day) than Māori (450 ± 56 min/day) or European (452 ± 75 min/day) women. Time spent in moderate PA and MVPA was significantly higher (*p* < 0.001) among European women than Māori or Pacific women (Table 2).

Reallocating sedentary behaviour to the activity of light intensity did not predict an improvement in either body composition (Figure 1) or metabolic markers (Figure 2) across any ethnic group. The model predicted that women of Māori descent would have decreased BF% (−14.84%) and insulin levels (−55.27%) if sedentary behaviour was replaced by moderate-intensity PA. Even greater improvements were predicted if MVPA replaced sedentary behaviour. While BF% (−16.16%) and insulin levels (−55.58%) improved, BMI (−15.33%), waist circumference (−9.98%), android fat% (−12.54%), gynoid fat% (−10.04%) and leptin (−43.86%) decreased when sedentary behaviour was switched to MVPA. Likewise, Pacific women were also predicted to benefit if sedentary behaviour was replaced with moderate intensity PA, particularly in decreased waist-to-hip ratio (6.40%) and android-to-gynoid fat ratio (19.48%). Waist-to-hip ratio continued to improve (5.30%) for Pacific women if sedentary behaviour was replaced with MVPA. For European women, no significant changes in body composition or metabolic markers were predicted with the reallocation of sedentary behaviour to PA at any intensity.

## 4. Discussion

The aim of this study was to investigate cross-sectional associations of increasing PA volume and intensity on body composition and metabolic health markers of New Zealand women, especially those from particularly vulnerable ethnic groups. An isotemporal substitution model was used to predict the effects on outcome variables of substituting sedentary behaviour with equal time engaged in PA of a different intensity. Isotemporal substitution analysis has not previously been used to predict changes in regional body composition, nor has it been conducted in women stratified by ethnicity. Thus, these findings among women with overweight/obesity, more robustly describe the importance between PA, and specifically PA intensity, on improved health indicators as related to a significant and vulnerable portion of the New Zealand female population.

The main finding was the differential association of increased PA on body composition and metabolic health variables among women of different ethnicities. An important and unexpected finding was the extensive benefits predicted for Māori women, but not for Pacific or European women. Māori women have an elevated risk of type 2 diabetes (RR 2.11) and cardiovascular disease (RR 2.1 for ischemic heart disease) and a higher prevalence of obesity (RR 1.66) than non-Māori women [7]. Hence, the improvements predicted in BMI, BF%, android fat%, waist circumference, and insulin for Māori women are particularly important and relevant for their long-term health. Just 30 min/day of sedentary time reallocated to MVPA predicted a 42.7% (7.0 mU/L) reduction in insulin, sufficient to return insulin concentrations to well within the healthy range. Exercise is known to improve insulin sensitivity, especially in obese populations [29,30,31], and improve metabolic flexibility and the ability of the body to switch between glucose and lipid metabolism depending on demands [32,33]. In this light, the body composition and insulin improvements predicted with increased PA present significant potential to enhance the long-term health of Māori women with overweight and obesity.

In contrast to predictions for Māori women, the lack of significant change predicted for Pacific women with increased PA was intriguing. Pacific women are at even greater risk of type 2 diabetes (RR 3.21 vs. non-Pacific women) than their Māori counterparts [7] and have an alarmingly high obesity prevalence (69.5%) [34], yet only waist-to-hip and android-to-gynoid fat ratios were predicted to change significantly. Nevertheless, the waist-to-hip ratio is an important indicator of central adiposity and disease risk [35,36], so any improvement in this measure is a positive health indicator for this group. The distinct differences between the ethnic groups in predicted responses to increased PA are unlikely to be related to different levels of PA since MVPA time did not differ significantly between Māori and Pacific women (*p* = 1.00). Percentages of fat and lean mass also did not differ significantly between the three groups (*p* > 0.15), even though BMI was higher among Māori and Pacific women than among European women; no body composition variables differed between Māori and Pacific women. Similarly, BMI but not BF% differed between Pacific and European adults or adolescents with obesity [4,37]. Ethnicity-dependent variations in body composition have also been reported in response to PA [38,39,40], although no studies were found to compare the current ethnic groups. Accelerometer cut-off points are population specific [41] and relationships between PA intensity and body composition may have varied had different accelerometer cut-off points been used, although it is not possible to determine from the current analysis. Furthermore, even though no differences in levels of PA were detected between Māori and Pacific women, the possibility of different levels of activity exists within each intensity range. In fact, previous analysis within the women’s EXPLORE study found that different classifications of sedentary behaviour (non-active vs. active) were significantly correlated with anthropometric and metabolic disease risk factors [42].

Across all ethnic groups, reallocation of sedentary time to moderate PA or MVPA was associated with only a few changes in metabolic health markers. Most notably, a 55% reduction in insulin was predicted among Māori women. Lower concentrations of insulin were also reported in adults following increased MVPA, and in those engaging in daily PA sufficient to meet PA guidelines, compared to less active adults [13,43,44]. The average concentration levels for biomarkers (except insulin) were within healthy ranges for this study. Therefore, predicted improvements may have been more pronounced in women with substantial disease risk profiles [45]. 

Within this subsample of women with overweight/obesity, 55% performed no vigorous PA during the seven-day accelerometer wear period. Hence, any increase in PA would likely have a marked effect on body composition. Indeed, improvements in body composition and metabolic markers (body mass; BMI; BF%; lean mass; waist circumference) were predicted with the reallocation of just 30 min/day from sedentary to more active behaviour. Potential exists for metabolic and body composition improvements of even greater magnitude if longer or more intense exercise was performed by these at-risk women. Significantly improved body composition (total, abdominal, subcutaneous, and visceral fat) was also reported among previously inactive women with obesity following a 14-week moderate-intensity exercise intervention [46]. In other exercise training studies of obese women, moderate intensity (continuous aerobic) exercise, and MVPA were more effective than high-intensity exercise at reducing total and android fat mass [47] and increasing lean mass [48].

Among the strengths of this study was the inclusion of total energy intake as a covariate in the isotemporal model, accounting for imbalances between energy intake and expenditure. The use of accelerometers to obtain objective PA data was a further strength. However, there are a number of limitations that must be considered. The accelerometer cut-points used may not be the most effective at distinguishing sedentary behaviour from light PA or at detecting differences in overall PA volume [41,49]. This limitation is not isolated to our findings; previous research in isotemporal substitution has reported conflicting results when replacing sedentary time with light PA [13,50,51]. Low participation rates in vigorous PA precluded analysis of this highest PA intensity; such analysis would have been interesting, had data been available. However, some individuals may not be capable, or willing, to exercise at an intensity sufficient to meet the vigorous threshold [52], and even an analysis of lower intensities provides valuable and compelling evidence for the beneficial effects of increased PA. The uneven and small size of the ethnic groups is a limitation of the study and was affected by difficulties in recruitment and obtaining valid accelerometer data from some ethnic groups. Special consideration should be given to participant recruitment and engagement when working with diverse ethnic groups [53]. Finally, as with any cross-sectional study, and given the theoretical nature of the time substitution in the current analysis, inference of causality is not possible.

## 5. Conclusions

In summary, important changes in body composition and metabolic health markers were predicted by reallocating sedentary time to moderate PA or MVPA, depending on the ethnicity of New Zealand women. This novel application of isotemporal substitution analysis has not previously been applied to groups stratified by ethnicity and has highlighted distinct differences in metabolic disease risks associated with these groups. Not unexpectedly, women with overweight/obesity were largely inactive (44% failed to meet national PA guidelines) and were predicted to benefit substantially from increased PA. Of particular significance were substantially improved body composition and metabolic health markers predicted for Māori women. The current findings demonstrate the varied impact of increased PA on women and translate into potential improvements in health status. Well-designed intervention studies are required to confirm these findings in order to generate more specific PA recommendations. Interventions and recommendations aimed at improving body composition and metabolic health outcomes should carefully consider the population under investigation, their metabolic health needs, and the specific activity intensity with which to replace sedentary time. For these specific ethnic communities, national and regional programmes such as Healthy Families (Healthy Families NZ, 2015) and Workplace Health (Auckland Regional Public Health Service, 2018) have been established to provide suggestions for simple and manageable initiatives to easily incorporate even small amounts of PA into normal daily life.

## Figures and Tables

**Figure 1 ijerph-19-08760-f001:**
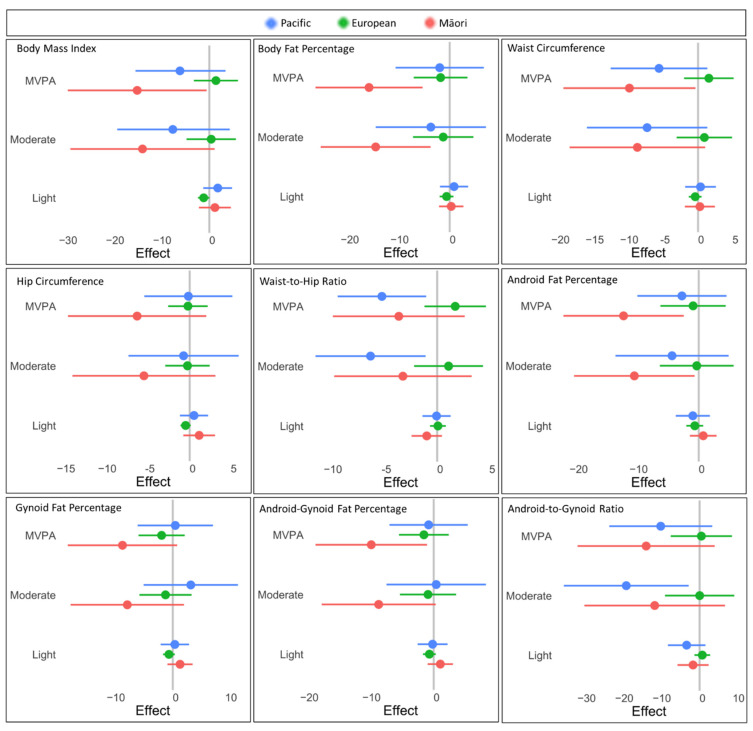
Reallocation effects across body composition variables. 95% confidence intervals for magnitude of predicted change in body composition variables when sedentary behaviour is reallocated for physical activity at light, moderate, and moderate-to-vigorous (MVPA) intensities. Participants are sub-stratified by ethnicity: Pacific (blue), European (green), and Māori (red).

**Figure 2 ijerph-19-08760-f002:**
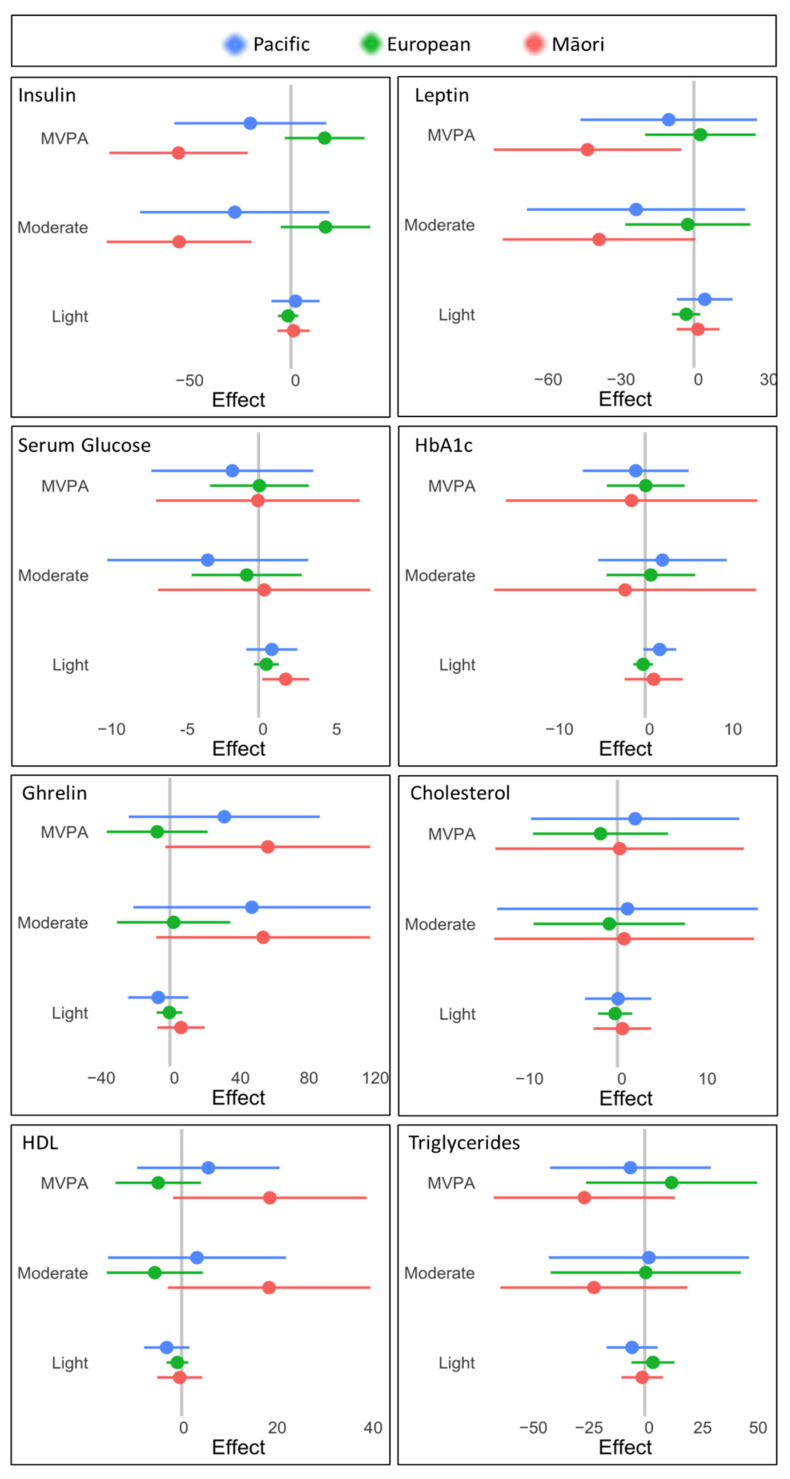
Reallocation effects across metabolic markers. 95% confidence intervals for magnitude of predicted change in metabolic markers when sedentary behaviour is reallocated for physical activity at light, moderate, and moderate-to-vigorous (MVPA) intensities. Participants are sub-stratified by ethnicity: Pacific (blue), European (green), and Māori (red).

**Table 1 ijerph-19-08760-t001:** Participant physical and metabolic health characteristics.

	Healthy Reference Range	Total (*n* = 175)	Māori (*n =* 37)	Pacific (*n =* 54)	European (*n =* 84)
*Physical characteristics*					
Age (y)		32.4 ± 8.6	31.9 ± 8.5	31.6 ± 9.2	33.1 ± 8.3
Body mass (kg)		86.4 ± 14.9	90.2 ± 17.0	93.1 ± 15.1	80.5 ± 11.2 ^‡^*
Height (cm)		166.5 ± 6.6	166.2 ± 4.3	167.4 ± 5.6	166.0 ± 6.9
BMI (kg/m^2^)	22.0–24.9	31.2 ± 5	32.6 ± 5.8	33.2 ± 5.2	29.2 ± 3.7 ^‡^*
Body fat (%)	18.0–29.9	39.7 ± 5.7	40.9 ± 5.8	40.2 ± 5.5	38.9 ± 5.7
Waist (cm)	<80.0	90.8 ± 10.4	93.0 ± 11.0	95.0 ± 10.8	87.1 ± 8.6 ^‡^*
Waist-to-hip ratio	<0.8	0.8 ± 0.1	0.80 ± 0.06	0.81 ± 0.06	0.79 ± 0.06 *
Android fat (%)		38.8 ± 5.2	40.5 (37.0, 44.5)	39.3 (37.1, 43.0)	38.3 (33.1, 41.4) ^‡^
Gynoid fat (%)		39.5 ± 4.2	39.3 (37.4, 42.1)	38.8 (37.1, 41.3)	39.9 (36.4, 43.2)
*Metabolic health markers*					
Systolic BP (mmHg)	<130	119 ± 11	119 ± 12	121 ± 10	117 ± 10
Diastolic BP (mmHg)	<80	75 ± 9	77 ± 10	76 ± 10	74 ± 8
HbA1c (mmol/mol)	<40.0	29.3 ± 3.9	30.4 ± 4.9	30.8 ± 3.2	27.8 ± 3.3 ^‡^*
Fasting glucose (mmol/L)	3.5–5.4	4.8 ± 0.4	4.86 ± 0.47	4.88 ± 0.42	4.71 ± 0.41
Insulin (mU/L)	3.0–25.0	15.1 ± 9.5	16.7 ± 8.3	19.6 ± 11.8	11.3 ± 6.3 ^‡^*
Cholesterol (mmol/L)	<5.0	4.6 ± 0.9	4.59 ± 0.71	4.25 ± 0.83	4.81 ± 0.95 *
HDL-c (mmol/L)	>1.0	1.4 ± 0.4	1.31 ± 0.32	1.36 ± 0.34	1.55 ± 0.35 ^‡^*
Triglycerides (mmol/L)	<2.0	1.1 ± 0.9	1.34 ± 0.64	1.03 ± 0.64	1.06 ± 1.03
LDL-c (mmol/L)	<3.4	2.7 ± 0.8	2.68 ± 0.71	2.42 ± 0.73	2.83 ± 0.92 *
Chol:HDL ratio	<4.5	3.4 ± 0.9	3.69 ± 0.92	3.26 ± 0.80	3.26 ± 0.93 ^‡^

Values are mean ± SD, or median (25th, 75th percentile). Abbreviations: BMI, body mass index; BP, blood pressure; HbA1c, glycosylated haemoglobin; HDL-C, high-density lipoprotein cholesterol; LDL-C, low-density lipoprotein cholesterol; Chol:HDL ratio, cholesterol to HDL-C ratio. ^‡^ significantly different to Māori (*p* < 0.05); * significantly different to Pacific (*p* < 0.05).

**Table 2 ijerph-19-08760-t002:** Physical activity and sedentary data for participants.

		Activity Intensity (min/Day)				
	*n*	Sedentary	Light	Moderate	Vigorous	MVPA
Māori	37	450 ± 56	328 ± 80	25 ± 16	0 (0, 2)	21 (16, 35)
Pacific	54	490 ± 67 ^‡^	306 ± 66	23 ± 16	0 (0, 1)	22 (11, 32)
European	84	452 ± 75 *	329 ± 82	34 ± 16 ^‡^*	1 (0, 3)	35 (23, 48) ^‡^*
*p*-value		0.004	0.202	<0.001	0.611	<0.001

Values are mean ± SD or median (25th, 75th percentile). Abbreviations: MVPA, moderate to vigorous physical activity. Ethnic groups: ^‡^ significantly different to Māori (*p* < 0.05); * significantly different to Pacific (*p* < 0.05).

## Data Availability

The data presented in this study are available on request from the first study author.

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
