# Peer review of "Replacing Sedentary Time with Physically Active Behaviour Predicts Improved Body Composition and Metabolic Health Outcomes"

_ijerph, 2022, doi:10.3390/ijerph19148760_

Round 1

Reviewer 1 Report

Title: Replacing sedentary time with physically activity behavior predicts improved body composition and metabolic health outcomes

Article Type: original scientific paper

Summary

This study has conducted to predict body composition and metabolic health marker changes from physical activity reallocation using isotemporal substitution analysis. Participants were 175 women with range of age between 16 to 45 years old. The results showed reallocating sedentary behavior with moderate intensity physically activity predicted Māori women would have improved body fat% (14.83%), android fat% (10.74%), and insulin levels (55.27%) while the model predicted Pacific women would have improved waist-to-hip (6.40%) and android-to-gynoid (19.48%) ratios. However, when reallocating sedentary behavior with moderate-vigorous intensity physically activity, the results predicted Māori women to have improved BMI (15.33%), waist circumference (9.98%), body fat% (16.16%), android fat% (12.54%), gynoid fat% (10.04%), insulin (55.58%) and leptin (43.86%) levels; and also, for Pacific women, improvement of waist-to-hip-ratio (5.30%) was reported. Overall, this study recommends that every level of physically activity that needed for improving of health, should reflect needs and current activity levels in every specific population.

Minor points and suggestions

·         Given that this study was performed on women, it is better to emphasize in the introduction to previous research on this population and also their differences between men and women on physical activity behavior.

·         It is better to give the reason for choosing women and their different races better in the introduction.

·         What is your reason for using Isotemporal substitution analysis? Please talk about it more in the introduction.

·         Do you have an ethic code for this study? please add it to the method section.

·         What was the reason for choosing this sample size for study?

  • Please add effect sizes for all comparisons in the results section following the statistical indicators.

Author Response

Comment 1: Given that this study was performed on women, it is better to emphasize in the introduction to previous research on this population and also their differences between men and women on physical activity behavior.

Author Reply 1: Thank you for this suggestion, we have addressed this through the revisions as outlined below.

Comment 2: It is better to give the reason for choosing women and their different races better in the introduction.

Author Reply 2: Gender and ethnicity differences to justify the study population have been added to paragraph 2 of the Introduction:

nevertheless, 36% of New Zealand women fail meet the physical activity guidelines (compared to 32% and 11%, respectively, among men) [7]. PA participation also differs by ethnicity and follows the ethnic pattern of obesity and its related diseases; 19% of Pacific women complete less than 30 minutes of PA per week whilst 15% of Māori and 13% of New Zealand European women [7].

Comment 3: What is your reason for using Isotemporal substitution analysis? Please talk about it more in the introduction.

Author Reply 3: The Introduction has been amended to further explain the use of isotemporal substitution analysis as follows:

Isotemporal substitution analysis is used to predict cardiometabolic health benefits of a more physically active lifestyle by reallocating time from one activity (e.g. sitting) to another (e.g. stepping) [12,13]. However, this technique has not been employed to examine changes in detailed body composition variables such as total and regional lean and fat mass with the reallocation of activities.

Comment 4: Do you have an ethic code for this study? please add it to the method section.

Author Reply 4: The ethics code is stated in the Method section as follows:

The study was approved by the Massey University Human Ethics Committee: Southern A, Reference No.13/13.

Comment 5: What was the reason for choosing this sample size for study?

Author Reply 5: The following sample size statement was added to section 2.6 Statistical analysis:

Power calculations were conducted to give a minimum of 37 participants in each group to detect an effect size of 0.58 with 80% power. At this number of participants, correlations of 0.44 were able to be detected with 80% power at a significance level of p < 0.05.

Comment 6:  Please add effect sizes for all comparisons in the results section following the statistical indicators.

Author Reply 6: Thank you for this suggestion, the graphs show effect sizes, as they display confidence intervals for the scaled magnitude of the effects in each case, rather than the degree of statistical significance.

Reviewer 2 Report

This is a clear and well-written manuscript evaluating the potential benefits of replacing sedentary behaviors with more physically active ones on body composition and metabolic health markers in different ethnic groups of New Zealand women. Content is original, and statistical analysis is correct, but the ethnic subsamples examined are small in size.

In general, the article is of interest to a wide audience,  stressing the importance of considering the different specificities of ethnic communities.

My suggestions mainly concern some methodological aspects and limitations to be clarified/improved:  

·       Line 85: Participants: It is unclear how the subsample of 206 women was selected.

·       Line 101: Why did 12 participants return with no data? Was there a malfunction of the instrument or did they not understand its use?

·       Line 112 and following: change "height" to "standing height" or "stature." Provide further indication on anthropometric methods used (e.g., head orientation in measuring stature).

·       Lines 130-132: Specify more precisely the method of measurement (e.g., lying down or sitting up) and specify whether the accepted pressure measurement (even though the process is automatic) was the lower of the two measurements.

·       line 279 and following: also report small ethnic subsample sizes as a limitation.

Minor concerns:

·       Captions in Figures 1 and 2 should be moved above the reference figure.

·       Line 320 and the following: Funding, Institutional Review Board Statement, and Informed Consent Statement have not been filled in.

Author Response

Reviewer 2

Comments to Author:  This is a clear and well-written manuscript evaluating the potential benefits of replacing sedentary behaviors with more physically active ones on body composition and metabolic health markers in different ethnic groups of New Zealand women. Content is original, and statistical analysis is correct, but the ethnic subsamples examined are small in size. In general, the article is of interest to a wide audience, stressing the importance of considering the different specificities of ethnic communities. My suggestions mainly concern some methodological aspects and limitations to be clarified/improved:  

Comment 1:  Line 85: Participants: It is unclear how the subsample of 206 women was selected.

Author Reply 1: Lines 85-88 were amended to clarify that the subsample of 206 women consisted of the overweight/obese women from the overall EXPLORE sample of 406.

Comment 2:  Line 101: Why did 12 participants return with no data? Was there a malfunction of the instrument or did they not understand its use?

Author Reply 2: Thank you for pointing out the lack of clarity here, we have updated this to now provide details for the missing data: - (device malfunction, n = 1; device not returned, n = 9; device not worn, n = 2).

Comment 3: Line 112 and following: change "height" to "standing height" or "stature." Provide further indication on anthropometric methods used (e.g., head orientation in measuring stature).

Author Reply 3: Thank you for this suggestion, “Height” has been changed throughout to “(standing) height” on Line 112. The authors do not see any need to specify “standing” in relation to height elsewhere. All anthropometric measurements were conducted according to the ISAK protocol, as referenced in Line 112-113 (reference 25 is (Marfell-Jones et al., 2006):

Anthropometric assessment was conducted using International Society for the Advancement of Kinanthropometry protocols [25].

Comment 4:  Lines 130-132: Specify more precisely the method of measurement (e.g., lying down or sitting up) and specify whether the accepted pressure measurement (even though the process is automatic) was the lower of the two measurements.

Author Reply 4: The procedure states that blood pressure was taken after 5 min of seated rest. The authors believe that this is sufficient to suggest that the measure was taken while seated. To clarify which of the two measurements was used “and the average of the two measurements was recorded” was added to Line 132.

Comment 5: line 279 and following: also report small ethnic subsample sizes as a limitation.

Author Reply 5: We have added “and small” to the limitations on Line 292. “The uneven and small size of the ethnic groups is a limitation of the study and was affected by difficulties in recruitment and obtaining valid accelerometer data from some ethnic groups.”

Comment 6:  Captions in Figures 1 and 2 should be moved above the reference figure.

Author Reply 6: We reviewed the figure captions, and they are provided above each figure.

Comment 7: Line 320 and the following: Funding, Institutional Review Board Statement, and Informed Consent Statement have not been filled in.

Author Reply 7: The statement regarding funding has been moved from the Acknowledgement section to the Funding section.
